# Bioinformatic Analysis of the BCL-xL/BCL2L1 Interactome in Patients with Pancreatic Cancer

**DOI:** 10.3390/medicina58111663

**Published:** 2022-11-17

**Authors:** Dimitrios E. Magouliotis, Anna P. Karamolegkou, Prokopis-Andreas Zotos, Evangelos Tatsios, Athina A. Samara, Dimitra Alexopoulou, Fani Koutsougianni, Nikos Sakellaridis, Dimitris Zacharoulis, Konstantinos Dimas

**Affiliations:** 1Department of Cardiothoracic Surgery, University of Thessaly, Biopolis, 41500 Larissa, Greece; 2Department of Surgery, University of Thessaly, Biopolis, 41500 Larissa, Greece; 3Department of Pharmacology, University of Thessaly, Biopolis, 41500 Larissa, Greece

**Keywords:** PDAC, pancreatic cancer, BCL-xL, bcl2l1, miRNA

## Abstract

*Objectives*: The aim of the present study was to analyze the differential gene expression of BCL-xL/BCL2L and the associated genetic, molecular, and biologic functions in pancreatic ductal adenocarcinoma (PDAC) by employing advanced bioinformatics to investigate potential candidate genes implicated in the pathogenesis of PDAC. *Materials and Methods*: Bioinformatic techniques were employed to build the gene network of BCL-xL, to assess the translational profile of BCL-xL in PDAC, assess its role in predicting PDAC, and investigate the associated biologic functions and the regulating miRNA families. *Results*: Microarray data extracted from one dataset was incorporated, including 130 samples (PDAC: 69; Control: 61). In addition, the expression level of BCL-xL was higher in PDAC compared to control samples (*p* < 0.001). Furthermore, BCL-xL demonstrated excellent discrimination (AUC: 0.83 [95% Confidence Intervals: 0.76, 0.90]; *p* < 0.001) and calibration (R squared: 0.31) traits for PDAC. A gene set enrichment analysis (GSEA) demonstrated the molecular functions and miRNA families (hsa-miR-4804-5p, hsa-miR-4776-5p, hsa-miR-6770-3p, hsa-miR-3619-3p, and hsa-miR-7152-3p) related to BCL-xL. *Conclusions*: The current findings unveil the biological implications of BCL-xL in PDAC and the related molecular functions and miRNA families.

## 1. Introduction

Pancreatic cancer (PC) currently represents one of the major causes of cancer-related mortality worldwide and is the fourth most frequent cause in the US [1,2]. Most of the patients that are diagnosed with PC present pancreatic ductal adenocarcinomas (PDAC) and the tumors are typically located in the head and uncinate process of the pancreas [3,4]. The disease is generally advanced during the diagnostic work-up and is related to a poor prognosis [5]. In this context, the current epidemiologic evidence shows that the 5-year overall survival rate of patients with PC is 11.5% [5]. In fact, depending on the independent characteristics of the tumor microenvironment, the clonal heterogeneity, and the degree of differentiation, the malignancy might present poorly formed to well-formed glands or infiltrating cells that form sheets [3,4]. In spite of the constant advances in PC research, the PC-related mortality rate continues to rise and it is estimated that by 2030 PC will represent the second most frequent cancer-related cause of mortality [6].

In this context, it has been documented that certain cancer activities, such as cell development, growth, invasion, and metastasis, highly depend on the metabolism rate along with the tumor microenvironment [7]. The BCL-2 protein family is characterized by their implication during the process of apoptosis [6]. BAK and BAX represent the two primary members of the family with pro-apoptotic attributes, which are essential during the execution phase of the apoptotic pathways [6,7], while there are other genes, like BCL-xL (gene/transcript name: BCL2L1), BCL2, BCL2A1, MCL1, and BCL-W, that are mainly associated with anti-apoptotic properties that promote the survival of the cells [8]. In fact, the distribution between the pro-apoptotic and anti-apoptotic members finally contributes to the death or survival of a cell. At present, the role of this BCL-2 family of proteins in PDAC is poorly understood. Despite the importance of BCL-xL in cell survival, the available evidence regarding its distorted expression/function in the context of PDAC is limited [9]. This information prompted us to perform a bioinformatic analysis regarding the role of BCL-xL in PDAC.

## 2. Materials and Methods

### 2.1. Nucleotide Sequential Analysis (NSA) of BCL-xL and Construction of Its Gene Network

The nucleotide sequence of the BCL-xL gene was searched and extracted from the Ensembl database. The next step was to employ the EMBOSS_CpGplot tool [10] to identify potential CpG islands by implementing the criteria by Takai and Jones [11]. Bioinformatic analyses of the BCL-xL-related interactome were performed to identify associated partners implicated in the PDAC pathogenesis. In fact, the BCL-xL interactome was obtained by employing the GeneMANIA platform (http://genemania.org/) (accessed on 10 April 2022) [12]. GeneMANIA is a platform that employs in silico techniques to predict the functions of more than one gene and to build a gene network based on specific gene ontology (GO) algorithms. Finally, the functions of the proteins associated with the identified genes were extracted from the portal GeneCards (http:// www.genecards.org/) (accessed on 10 April 2022), which is a platform containing information regarding the human genome.

### 2.2. In Silico Assessment of the Expression Levels of the BCL-xL Gene

The PubMed Gene Expression Omnibus (GEO) database (https://www.ncbi.nlm.nih.gov/gds) (accessed on 10 April 2022) was employed to assess the expression levels of the BCL-xL gene in PDAC as compared with healthy tissues. PubMed GEO is a publicly available database containing gene expression datasets. A systematic search was performed in PubMed GEO using the following keywords: “pancreatic cancer”, “pancreatic ductal adenocarcinoma”, and “pdac”, using “Homo Sapiens” as the tissue donator species filter. The database search was conducted until April 2022 by two independent researchers (DEM, APK). The level of agreement between the two reviewers was estimated by employing the kappa coefficient test of agreement.

The expression profile was estimated using a unique PDAC microarray dataset (GSE62452), including a total of 130 samples (PDAC: 69 PDAC; adjacent non-tumor tissue control: 61 samples). The datasheet was obtained by the [HuEx-1_0-st] Affymetrix Human Exon 1.0 ST Array platform and the gene expression data were log-transformed; moreover, we examined the null hypothesis, which suggested that the gene expression levels between the cancer and healthy tissue samples were comparable. Genes were considered as significantly upregulated or downregulated when *p* < 0.05.

### 2.3. Validation of Discrimination and Calibration Traits of BCL-xL in the Context of PDAC

We evaluated the discrimination and calibration of BCL-xL in the context of PDAC diagnosis. We evaluated discrimination by constructing receiver-operating characteristic (ROC) curves and by estimating the area under curve (AUC). The AUC was determined by estimating the 95% confidence intervals (95% CI) and compared by employing non-parametric paired tests, as described by DeLong et al. [13]. The model discrimination was defined as either poor, fair, or excellent when the AUC was <0.70, 0.70–0.79, and 0.80–1.00, respectively. Furthermore, we assessed the calibration traits for each model by estimating the predicted BCL-xL incidence (expected) and then compared it with the true BCL-xL presentation (observed). The observed/expected ratio of 1 represented perfect accuracy, a ratio <1 indicated overprediction of BCL-xL rate, and a ratio of >1 indicated underestimation. In addition, we further assessed calibration of the model by employing the Hosmer–Lemeshow (H-L) goodness of fit test and defined a lack of fit as a *p* ≤ 0.05 [14]. Finally, we employed the chi-squared test to compare the observed and expected outcomes of all patients.

### 2.4. Gene Set Enrichment Analysis (GSEA) Regarding the Molecular Functions and Regulating miRNAs of BCL-xL

The GSEA of Gene Ontologies (GO) was performed using the portal ToppFun/ToppGene (https://toppgene.cchmc.org/) (accessed on 10 April 2022). ToppFun reports all the important enrichments of the biologic functions and regulating miRNAs using data derived from transcriptome, proteome, phenotype, pharmacome, and bibliome [15]. The significant enrichments were further assessed by estimating their false discovery rates (FDR). All the analyses were performed in May 2022.

### 2.5. Statistical Analysis

To analyze all the extracted data, we employed the software GraphPad Prism 9.4.1 (GraphPad Software, San Diego, CA, USA). We examined the normal distribution of the data by employing the D’Agostino and Pearson Omnibus normality test. Furthermore, we compared the gene expression levels by using a two-tailed unpaired t-test for parametric data or a Mann–Whitney U-test for non-parametric data. All *p*-values were corrected for multiple comparisons by calculating the Q statistic (Benjamini–Hochberg test), with a threshold for statistical significance of Q < 0.05. In addition, correlations were evaluated by estimating the Pearson or Spearman’s rank (ρ) correlation coefficients for parametric or non-parametric data, respectively. Finally, we performed a Deming regression analysis to evaluate important relationships among differentiated genes.

## 3. Results

### 3.1. NSA of BCL-xL and Building of Its Interactome

The trial flow of the present study is shown in Figure 1. The NSA identified a unique CpG island [length: 513 (1, 513)] associated with BCL-xL and based on the following criteria: percent C + percent G >50.00; observed/expected ratio >0.60; and length >200 (Figure 2).

The members of the BCL-xL interactome that were provided from the GeneMania portal are shown in Table 1, Figure 3. In total, 21 interacting proteins were identified by constructing the BCL-xL interactome in homo sapiens.

### 3.2. In Silico Assessment of the Expression Levels of the BCL-xL Network

A unique PubMed GEO PDAC microarray dataset was identified (GSE62452), which included a total of 130 samples (PDAC: 69; adjacent non-tumor tissue control: 61), with a substantial level of agreement between the reviewers (Kappa = 0.651; 95% CI: 0.006, 1.000). According to our outcomes, BCL-xL is upregulated in PDAC (*p* < 0.001), as highlighted in Figure 4.

### 3.3. Validation of the Discrimination and Calibration Traits of BCL-xL in the Context of PDAC

BCL-xL provided an excellent discrimination level (AUC: 0.83 [95% Confidence Intervals: 0.76, 0.90]; *p* < 0.001). The ROC curve is depicted in Figure 5. BCL-xL also provided an adequate estimation of the PDAC incidence (O: E = 1). Furthermore, the goodness of fit test was positive and the Tjur’s R squared test was low (R squared = 0.31). Consequently, BCL-xL provided excellent discrimination and calibration traits.

### 3.4. GSEA of BCL-xL

BCL-xL underwent GSEA (Figure 6). The five most significantly enriched GO terms for molecular functions are presented in Table 2, along with the regulating miRNAs. Regulation of BH, BH3, and MDM2/MDM4 family protein binding activities, along with cysteine-type endopeptidase inhibitor activity involved in apoptotic processes, represented the most significant molecular functions relevant to BCL-xL. Finally, the GSEA proposed that the members of the hsa-miR-4804-5p, hsa-miR-6770-3p, hsa-miR-3619-3p, hsa-miR-4776-5p, and hsa-miR-7152-3p families play a significant role in the regulation of BCL-xL-associated pathways.

## 4. Discussion

BCL-xL and BCL2 have been reported to play a crucial role in the pancreas during the state of pancreatitis [16]. Current evidence has demonstrated that BCL-xL regulates the apoptotic pathway in pancreatic tissue through downstream apoptotic cleaving of caspases 3 and 313 and the consequent caspase cascade [17,18,19]. Intriguingly, the upregulation of BCL-xL expression and function seems to result in a significant increase in pancreatic transcription and the levels of proteinic expression, while BCL-xL plays a crucial role in protection from apoptotic stimuli of mature B-cells [20].

The present study underscores the implication of BCL-xL in pancreatic tumor biology. We herein assessed the gene expression profile of the BCL-xL in patients with PDAC by using in silico techniques and incorporating data provided by a PDAC microarray dataset, thus providing enhanced accuracy at the molecular level. A total of 130 samples were incorporated and analyzed in the current analysis. The first result of our analysis is that BCL-xL is overexpressed in PDAC as compared to adjacent normal (non-cancerous) tissue. This outcome is in accordance with previously published evidence on the potential role of BCL-xL in a study performed by immunohistochemistry in nine human PanINs and PDAC [9]. We also assessed the discrimination and calibration traits of BCL-xL in PDAC. In this context, we found that BCL-xL is associated with excellent discrimination and calibration traits.

Moreover, we herein found that there is a unique nucleotide CpG island associated with the BCL-xL gene, and we constructed the interactome of BCL-xL. This evidence provided for the first time by the present study should, however, be further investigated to uncover the role of BCL-xL in the pathogenesis of pancreatic ductal adenocarcinoma, along with its potential role as a prognostic marker and a potential drug target—of course.

In our study, we further implemented miRNA datasets from *Homo sapiens* to establish and better unveil the BCL-xL implication in PDAC biology. Our findings highlight some miRNA families related to BCL-xL. Indeed, we found, and reported for the first time, the BCL-xL-associated miRNAs, hsa-miR-4804-5p, hsa-miR-6770-3p, hsa-miR-3619-3p, hsa-miR-4776-5p, and hsa-miR-7152-3p, which may be regulators of BCL-xL. This newly acquired evidence should be investigated to a greater extent to increase the level of knowledge regarding PDAC pathogenesis and treatment, along with enhancing our treatment strategy options.

The dysregulation of BCL-xL function is, interestingly, also reported to indirectly affect metabolic processes in the residual pancreatic tissue. In fact, there is certain evidence highlighting the role of BCL-xL in cellular metabolism, differentiation, and development, thus affecting pancreatic function and development [21]. According to our enrichment analysis, BCL-xL is associated with BH, BH3, and MDM2/MDM4 family protein binding activities, along with cysteine-type endopeptidase inhibitor activity. These findings are in accordance with previous evidence indicating that BCL-xL is involved in the energetic capacity necessary for cell survival and apoptotic pathways [22]. Thus, regulation of BCL-xL expression in the different stages of PC pathogenesis could potentially represent an interesting field of further study.

The present study is associated with certain limitations: (a) the relatively small number of samples, (b) the lack of available patients’ baseline data to perform further multivariate analyses, and (c) the lack of data on mutations and alterations that could provide stronger evidence than the gene expression data. Nevertheless, there are certain strengths associated with the current analysis, which are (a) the well-designed protocol, (b) the homogeneity of tissue-treating regarding the included tissue samples, (c) the use of a unique Affymetrix chip in all samples, and (d) the performance of GSEA regarding significantly differentiated genes, which demonstrated the miRNA families and biologic functions that are associated with the identified biomarkers.

## 5. Conclusions

In the current in silico study, we highlight the potential role of BCL-xL in PDAC biology. We found that BCL-xL is overexpressed in pancreatic ductal adenocarcinoma with excellent discrimination and calibration traits. Furthermore, the present analysis provides evidence regarding the anti-apoptotic traits of this protein, along with its relevance with BH, BH3, and MDM2/MDM4 family protein binding activities, and with cysteine-type endopeptidase inhibitor activity. Finally, our findings highlight miRNAs related to BCL-xL and the potential role of methylation in the function of BCL-xL in PDAC. Since this is an in silico analysis, deeper research with greater clarity in important endpoints, such as a multiple regression analysis, will fully uncover the potential benefit from our outcomes regarding the PDAC diagnostic work-up and treatment strategy.

## Figures and Tables

**Figure 1 medicina-58-01663-f001:**
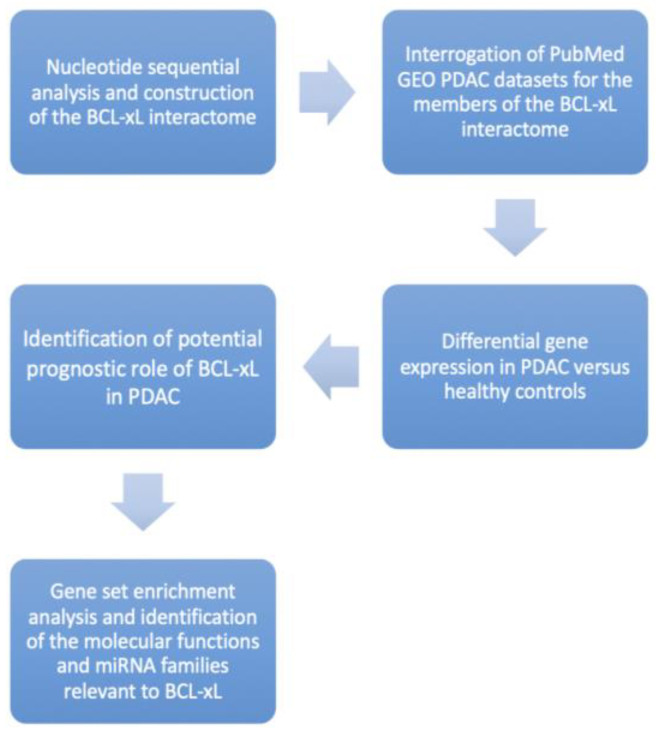
Study Trial Flow demonstrating our research protocol step-by-step.

**Figure 2 medicina-58-01663-f002:**
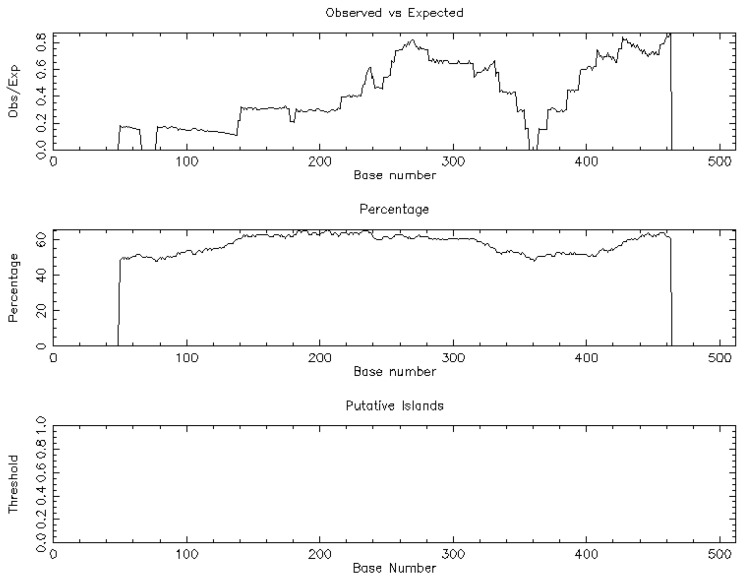
The nucleotide sequential analysis demonstrated one CpG island related to BCL-xL.

**Figure 3 medicina-58-01663-f003:**
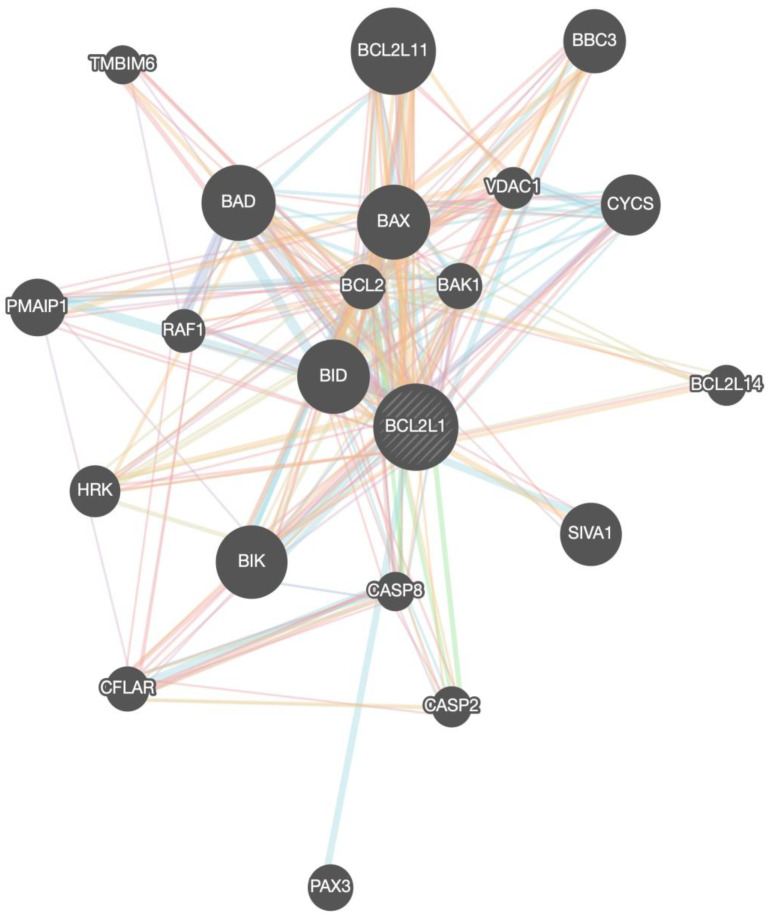
BCL-xL/BCL2L1 gene network (interactome) with all the associated genes.

**Figure 4 medicina-58-01663-f004:**
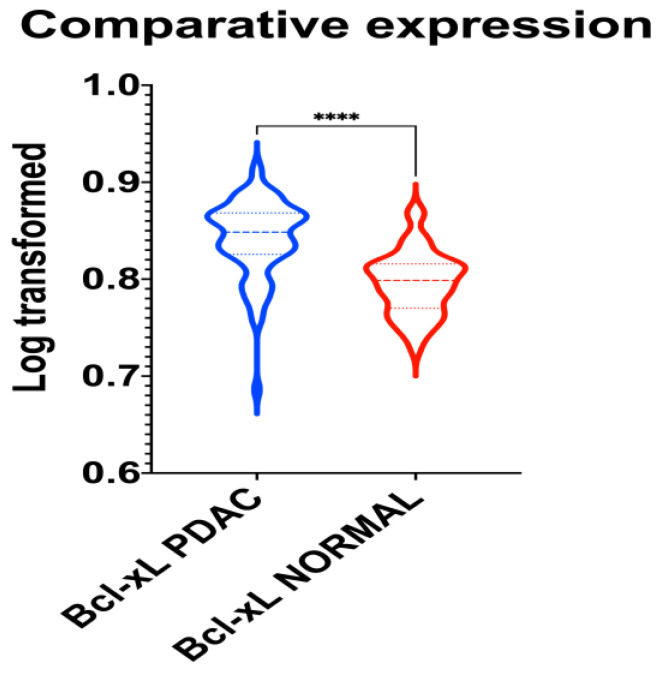
Violin plot demonstrating the differential gene expression of BCL-xL in normal and PDAC tissue samples. ****: *p* < 0.001.

**Figure 5 medicina-58-01663-f005:**
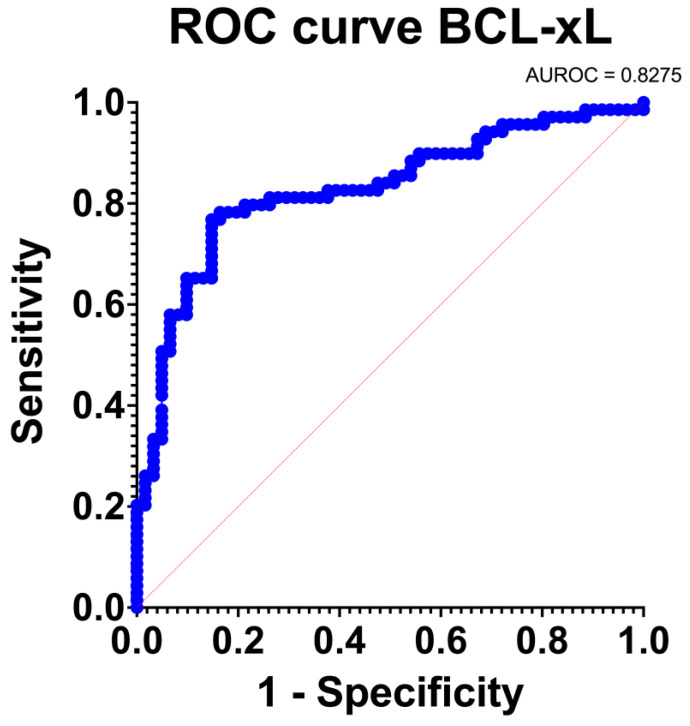
Receiver-Operating Characteristic (ROC) curve regarding the sensitivity.

**Figure 6 medicina-58-01663-f006:**
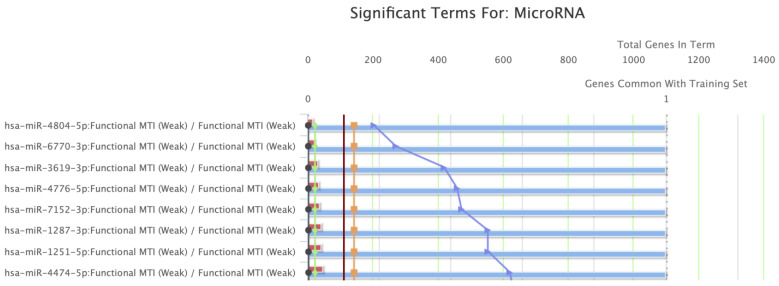
Gene Set Enrichment Analysis (GSEA) indicated miRNA families that are important regulators of BCL-xL.

**Table 1 medicina-58-01663-t001:** Gene symbols and description of the BCL-xL interactome members.

Gene Symbol	Gene Description
BCL2L1	BCL2 Like 1
BCL2	BCL2 Apoptosis Regulator
BID	BH3 Interacting Domain Death Agonist
BAK1	BCL2 Antagonist/Killer 1
BAX	BCL2 Associated X, Apoptosis Regulator
BAD	BCL2 Associated Agonist Of Cell Death
RAF1	Raf-1 Proto-Oncogene, Serine/Threonine Kinase
BCL2L11	BCL2 Like 11
BIK	BCL2 Interacting Killer
CASP8	Caspase 8
CASP2	Caspase 2
PAX3	Paired Box 3
CFLAR	CASP8 And FADD Like Apoptosis Regulator
HRK	Harakiri, BCL2 Interacting Protein
PMAIP1	Phorbol-12-Myristate-13-Acetate-Induced Protein 1
TMBIM6	Transmembrane BAX Inhibitor Motif Containing 6
BBC3	BCL2 Binding Component 3
VDAC1	Voltage Dependent Anion Channel 1
CYCS	Cytochrome C, Somatic
SIVA1	SIVA1 Apoptosis Inducing Factor
BCL2L14	BCL2 Like 14

**Table 2 medicina-58-01663-t002:** Enrichment analysis of gene ontologies for the prognostic factors. Top five relevant molecular functions and regulating miRNA families are presented.

	**ID**	**Name**	**Source**	***p*-Value**	**FDR B&H**	**FDR B&Y**	**Bonferroni**
1	GO:0051434	BH3 domain binding	ToppFun	2.598 × 10^−4^	4.832 × 10^−3^	1.946 × 10^−2^	8.053 × 10^−3^
2	GO:0097371	MDM2/MDM4 family protein binding	ToppFun	4.676 × 10^−4^	4.832 × 10^−3^	1.946 × 10^−2^	1.450 × 10^−2^
3	GO:0070513	death domain binding	ToppFun	5.195 × 10^−4^	4.832 × 10^−3^	1.946 × 10^−2^	1.611 × 10^−2^
4	GO:0051400	BH domain binding	ToppFun	6.234 × 10^−4^	4.832 × 10^−3^	1.946 × 10^−2^	1.933 × 10^−2^
5	GO:0043027	cysteine-type endopeptidase inhibitor activity involved in apoptotic process	ToppFun	1.351 × 10^−3^	8.375 × 10^−3^	3.373 × 10^−2^	4.187 × 10^−2^
**Regulating miRNA Families**							
	**Name**		**Source**	***p*-Value**	**FDR B&H**	**FDR B&Y**	**Bonferroni**
1	hsa-miR-4804-5p		miRTarBase	2.076 × 10^−4^	9.614 × 10^−3^	6.412 × 10^−2^	9.177 × 10^−2^
2	hsa-miR-6770-3p		miRTarBase	2.768 × 10^−4^	9.614 × 10^−3^	6.412 × 10^−2^	1.224 × 10^−1^
3	hsa-miR-3619-3p		miRTarBase	4.291 × 10^−4^	9.614 × 10^−3^	6.412 × 10^−2^	1.897 × 10^−1^
4	hsa-miR-4776-5p		miRTarBase	4.706 × 10^−4^	9.614 × 10^−3^	6.412 × 10^−2^	2.080 × 10^−1^
5	hsa-miR-7152-3p		miRTarBase	4.845 × 10^−4^	9.614 × 10^−3^	6.412 × 10^−2^	2.141 × 10^−1^

## Data Availability

The data supporting the present study is available in the PubMed GEO (https://www.ncbi.nlm.nih.gov/gds), reference number GSE62452 (Accessed on 10 April 2022).

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
