# Peer review of "Bioinformatic Analysis of the BCL-xL/BCL2L1 Interactome in Patients with Pancreatic Cancer"

_medicina, 2022, doi:10.3390/medicina58111663_

Round 1

Reviewer 1 Report

The manuscript entitled `In-depth bioinformatic study of the BCL-xL/BCL2L1 gene in patients with pancreatic cancer` by Magouliotis et al. proposes the in silico analysis of BCL-xL in patients with pancreatic cancer from public databases.   The manuscript is well designed and written. The authors find an interesting potential role of BCL-xL as a predictor and/or biomarker for PDAC.   I have no concerns about the work and, beyond the interesting results, I highlight the clear description of the methodology used including the workflow diagram in Fig.1 and the importance for readers of the discussion`s last paragraph mentioning the limitations and advantages of the results.

Author Response

Reviewer #1, Comment #1: “The manuscript entitled `In-depth bioinformatic study of the BCL-xL/BCL2L1 gene in patients with pancreatic cancer` by Magouliotis et al. proposes the in-silico analysis of BCL-xL in patients with pancreatic cancer from public databases.   The manuscript is well designed and written. The authors find an interesting potential role of BCL-xL as a predictor and/or biomarker for PDAC.   I have no concerns about the work, and, beyond the interesting results, I highlight the clear description of the methodology used including the workflow diagram in Fig.1 and the importance for readers of the discussion`s last paragraph mentioning the limitations and advantages of the results.”

Response: We would like to deeply thank the reviewer for his/her kind and encouraging comments.

Reviewer 2 Report

Major comments:

1.       This work has been previously published by the same authors elsewhere (https://www.hpbonline.org/article/S1365-182X(21)00926-6/fulltext). What is the new data or knowledge which the authors are presenting which would add novelty and interest to the biomedical or PDAC community? Also other research have already illustrated the role and impact of Bcl-xL in pancreatic cancer (https://www.ncbi.nlm.nih.gov/pmc/articles/PMC5604117/).

2.       The authors have analyzed data from other publicly available sources. While they correlate the over-expression of BCL-xL with PDAC as compared to control, there is no correlation made with the methylation data. The authors must correlate their CpG methylation data analysis with the overexpression data analysis, in other words, does the tissues obtained from controls show high methylation as compared to PDAC patients?

Minor Comments:1. Authors need to update the statistics – “In fact, the 5-year survival rate of pancreatic cancer is approximately 6%”.

Author Response

Comment #1:   This work has been previously published by the same authors elsewhere (https://www.hpbonline.org/article/S1365-182X(21)00926-6/fulltext). What is the new data or knowledge which the authors are presenting which would add novelty and interest to the biomedical or PDAC community? Also, other research has already illustrated the role and impact of Bcl-xL in pancreatic cancer (https://www.ncbi.nlm.nih.gov/pmc/articles/PMC5604117/).”

Response:

Regarding this first comment that this work has been published elsewhere we ensure the reviewer that this is not the case. The DOI provided by the reviewer corresponds to a published conference abstract for a poster presented during the 14th Congress of the European-African Hepato-Pancreato-Biliary Association (E-AHPBA), 15-17 September 2021, Bilbao. Virtual Congress. This is the reason that it is included in a Supplement edition of the journal.

The second study that the reviewer brings into our attention, for which we thank him/her and we apologize for missing it, is indeed a study on the potential role of BCL-xL in PDAC, the findings of which are actually in agreement with our results. A comment on this work has now been added to the Discussion section.  This though is a study mainly on mice with only a few human samples (only 9 PDAC samples being analyze with IHC and 7 controls originating from other pancreatic cancer tumor types).  Our study is based on a sample, that although relatively small is much bigger as compared to the sample size of this study (in our study n=130, 69 PDAC, 61 controls) and we also provide not only information on the importance of BCL-xL in PDAC but also a gene set enrichment analysis (GSEA) regarding the molecular functions and regulating miRNAs of BCL-xL. We also assessed the discrimination and calibration traits of BCL-xL in PDAC, we report that there is a unique nucleotide CpG island associated with the BCL-xL gene (that puts the basis for further exploitation), and we finally, constructed the interactome of BCL-xL.

 Comment #2: The authors have analyzed data from other publicly available sources. While they correlate the over-expression of BCL-xL with PDAC as compared to control, there is no correlation made with the methylation data. The authors must correlate their CpG methylation data analysis with the overexpression data analysis, in other words, does the tissues obtained from controls show high methylation as compared to PDAC patients?”

Response: We thank the reviewer for this valuable comment. Indeed, one of our original aims was to analyze methylation data as well. However, we could not proceed with this analysis in-silico due to limited available data. Unfortunately, our notion is that the only way to address this important question will be by a direct methylation analysis of tumors. Indeed, we have already decided to start building a biobank with patients’ tumors to proceed with this analysis following this work in a future study.

Comment #3: 1. Authors need to update the statistics – “In fact, the 5-year survival rate of pancreatic cancer is approximately 6%”.”

Response: Thank you for your valuable comment. We updated the statistics based on the SEER database evidence.

Reviewer 3 Report

The BCL-2 family of proteins are notable for their involvement in the process apoptosis [

Rephrase. E.g. apoptotic process

Despite the importance of BCL-xL in cell survival, there is only limited evidence regarding its altered expression and function in the context of PDAC.

This requires a reference

TAA abbreviation should be explained in its first use.

The BCL-xL gene network was produced

It would be better to say The BCL-xL gene network was obtained

histological analyses were obtained from resected pancreatic tumor

histological analyses were obtained from resected pancreatic tumors

The overexpression of BCL-xL might result in enhanced tumor growth through the dysregulated apoptotic pathways. Consequently, pancreatic cancer tumor growth is directly related to anti-apoptotic attributes of BCL-xL. In this context, we found that BCL-xL is associated with excellent calibration and discrimination traits, thus being proposed as a potential biomarker for early diagnosis.

I do not think the concepts above are proven by the studies performed for the paper.

Growth and apoptosis can be independent issues.

The early biomarker concept cannot be accepted at face value. Where the samples obtained from "early" tumors? We do not know. A marker obtained from a tissue specimen is useless for bedside diagnosis. Therefore we are talking here of a marker that is not aimed to diagnosis. Regarding prognosis, there is one thing that pancreatic tumors do not need: prognostic markers. On the other hand, PDAC lacks early diagnostic markers useful for screening. BCL-xL is not a marker for screening.

Furthermore, to show that BCL-xL can become a marker, the authors need to demonstrate that it is not expressed in any other tumors. this is lacking in the paper.

I would suggest the authors to leave out the marker issue.

Interestingly, the dysregulation of BCL-xL function also indirectly affected metabolic processes in the residual pancreatic tissue possibly due to the loss of cells that were more dependent on BCL-xL for survival.

This sentence is not clear and it is highly speculative unless references can be provided.

Modulation of the expression level of BCL-xL during pancreatic tumor specification could possibly be a means to regulate tumor growth and metastasis.

Highly speculative unless references are provided.

This paper shows three things:

1)      That BCL-xL expression is increased in pancreatic ductal adenocarcinoma (PDAC).

2)      That this antiapoptotic protein is associated with BH, BH3, MDM2/MDM4 family protein binding activities, and  with cysteine-type endopeptidase inhibitor activity.  

3)      That MiRNAs hsa-miR-4804-5p, hsa-miR-6770-3p, hsa-miR-3619-3p, hsa-miR-4776-5p, and hsa-miR-7152-3p are  regulators of BCLxL.

This paper does not show that

1)      BCL-xL is an early marker of PDAC.

2)      BCL-xL is a prognostic marker of PDAC

3)      BCL-xL is a marker of anything.

4)      BCL-xL is actually a driver of the tumor or that it regulates its metabolism.

Authors should restrict their report to their findings and omit speculative considerations not based on any evidence.

Author Response

Comment #1: “The BCL-2 family of proteins are notable for their involvement in the process apoptosis [

Rephrase. E.g. apoptotic process”

Response: Thank you for your comment. We rephrased it.

Reviewer #3, Comment #2: “Despite the importance of BCL-xL in cell survival, there is only limited evidence regarding its altered expression and function in the context of PDAC.

This requires a reference”

Response: Thank you for your comment. We provided a reference as you suggested.

Reviewer #3, Comment #3: “TAA abbreviation should be explained in its first use.”

Response: Thank you for your comment. We deleted this abbreviation.

Reviewer #3, Comment #4: “The BCL-xL gene network was produced

It would be better to say The BCL-xL gene network was obtained”

Response: Thank you for your comment. We rephrased it.

Reviewer #3, Comment #5: “histological analyses were obtained from resected pancreatic tumor

histological analyses were obtained from resected pancreatic tumors”

Response: Thank you for your comment. We rephrased it.

Reviewer #3, Comment #6: “The overexpression of BCL-xL might result in enhanced tumor growth through the dysregulated apoptotic pathways. Consequently, pancreatic cancer tumor growth is directly related to anti-apoptotic attributes of BCL-xL. In this context, we found that BCL-xL is associated with excellent calibration and discrimination traits, thus being proposed as a potential biomarker for early diagnosis.

I do not think the concepts above are proven by the studies performed for the paper.”

Response: Thank you for your comments. We rephrased this paragraph.

Reviewer #3, Comment #7: “The early biomarker concept cannot be accepted at face value. Where the samples obtained from "early" tumors? We do not know. A marker obtained from a tissue specimen is useless for bedside diagnosis. Therefore we are talking here of a marker that is not aimed to diagnosis. Regarding prognosis, there is one thing that pancreatic tumors do not need: prognostic markers. On the other hand, PDAC lacks early diagnostic markers useful for screening. BCL-xL is not a marker for screening.

Furthermore, to show that BCL-xL can become a marker, the authors need to demonstrate that it is not expressed in any other tumors. this is lacking in the paper.

I would suggest the authors to leave out the marker issue.”

Response: Thank you for your comments. We rephrased it and leaved out the marker issue as you suggested.

Reviewer #3, Comment #8: “Interestingly, the dysregulation of BCL-xL function also indirectly affected metabolic processes in the residual pancreatic tissue possibly due to the loss of cells that were more dependent on BCL-xL for survival.

This sentence is not clear and it is highly speculative unless references can be provided.”

Response: Thank you for your comments. We rephrased this sentence.

Reviewer #3, Comment #9: “Modulation of the expression level of BCL-xL during pancreatic tumor specification could possibly be a means to regulate tumor growth and metastasis.

Highly speculative unless references are provided.”

Response: Thank you for your comments. We rephrased this sentence.

Reviewer #3, Comment #10: “This paper shows three things:

1)      That BCL-xL expression is increased in pancreatic ductal adenocarcinoma (PDAC).

2)      That this antiapoptotic protein is associated with BH, BH3, MDM2/MDM4 family protein binding activities, and with cysteine-type endopeptidase inhibitor activity. 

3)      That MiRNAs hsa-miR-4804-5p, hsa-miR-6770-3p, hsa-miR-3619-3p, hsa-miR-4776-5p, and hsa-miR-7152-3p are  regulators of BCLxL.

This paper does not show that

1)      BCL-xL is an early marker of PDAC.

2)      BCL-xL is a prognostic marker of PDAC

3)      BCL-xL is a marker of anything.

4)      BCL-xL is actually a driver of the tumor or that it regulates its metabolism.

Authors should restrict their report to their findings and omit speculative considerations not based on any evidence.”

Response: Thank you for your comments. We rephrased the text according to your suggestions.

Round 2

Reviewer 2 Report

2.       The authors have analyzed data from other publicly available sources. While they correlate the over-expression of BCL-xL with PDAC as compared to control, there is no correlation made with the methylation data. The authors must correlate their CpG methylation data analysis with the overexpression data analysis, in other words, does the tissues obtained from controls show high methylation as compared to PDAC patients?

Reviewer 3 Report

I think that now the paper is in conditions for publication.